# Iterative Upgrading of Small Molecular Tyrosine Kinase Inhibitors for EGFR Mutation in NSCLC: Necessity and Perspective

**DOI:** 10.3390/pharmaceutics13091500

**Published:** 2021-09-18

**Authors:** Jing Zhu, Qian Yang, Weiguo Xu

**Affiliations:** 1Respiratory and Critical Care Medicine, Mianyang Central Hospital, Mianyang 621000, China; zhujing_1126@126.com; 2School of Medicine, University of Electronic Science and Technology of China, Mianyang 621000, China; 3Sichuan Province College Key Laboratory of Structure-Specific Small Molecule Drugs, School of Pharmacy, Chengdu Medical College, No. 783, Xindu Avenue, Xindu District, Chengdu 610500, China

**Keywords:** epidermal growth factor receptor tyrosine kinase inhibitors, EGFR mutations, molecular targeted therapy, non-small cell lung cancer, resistance mechanism

## Abstract

Molecular targeted therapy has been reported to have fewer adverse effects, and offer a more convenient route of administration, compared with conventional chemotherapy. With the development of sequencing technology, and research on the molecular biology of lung cancer, especially whole-genome information on non-small cell lung cancer (NSCLC), various therapeutic targets have been unveiled. Among the NSCLC-driving gene mutations, epidermal growth factor receptor (EGFR) mutations are the most common, and approximately 10% of Caucasian, and more than 50% of Asian, NSCLC patients have been found to have sensitive EGFR mutations. A variety of targeted therapeutic agents for EGFR mutations have been approved for clinical applications, or are undergoing clinical trials around the world. This review focuses on: the indications of approved small molecular kinase inhibitors for EGFR mutation-positive NSCLC; the mechanisms of drug resistance and the corresponding therapeutic strategies; the principles of reasonable and precision molecular structure; and the drug development discoveries of next-generation inhibitors for EGFR.

## 1. Introduction

Lung cancer is a serious threat to human health [1]. A global cancer report published by the World Health Organization (WHO) in 2020 reports that lung cancer is still the most common and fatal cancer, and it affects both men and women [2]. Lung cancer patients not only experience great psychological pressure, but also suffer severe pain due to the disease. Moreover, the high cost of treatment presents a great burden to both individuals and society [3].

The goal of advanced lung cancer treatment is to prolong the overall survival time of patients and to improve their quality of life [4]. Chemotherapy is one approach that kills cancer cells, and it can be administered orally or by injection. Platinum-based chemotherapy in combination with other cytotoxic drugs for 4–6 cycles is the current routine treatment. However, chemotherapy is not recommended for elderly patients with weak conditions, cachexia, serious dysfunction of the heart, liver, or kidney, or poor bone marrow function. With the advanced improvement of sequencing technology and in-depth research on the molecular biology of lung cancer, especially whole-genome sequencing, targeted therapy for solid tumors has rapidly developed. This has brought good news for, especially advanced, non-small-cell lung cancer (NSCLC) patients.

With the advancements in modern medicine, the diagnosis of, and therapy for, NSCLC have entered the era of “precision medicine”, facilitating more accurate diagnosis and treatment. According to the latest National Comprehensive Cancer Network (NCCN) guidelines for NSCLC (3rd Edition, 2021), the genes with driver mutations include: epidermal growth factor receptor (*EGFR*); anaplastic lymphoma kinase (*ALK*), c-ros oncogene 1 receptor tyrosine kinase (*ROS1*); human epidermal growth factor receptor 2 (HER2); mesenchymal to epithelial transition factor (MET); v-raf murine sarcoma viral oncogene homolog B1 (*BRAF*); Kirsten rat sarcoma (*KRAS*); rearrangement during transfection (*RET*); and neurotrophic tyrosine receptor kinase (*NTRK*) [5]. Individualized molecular targeted therapy for driver genes has been reported to block the key signaling pathway of tumor cell growth and proliferation, inhibit tumor cell proliferation, and selectively kill tumor cells by applying highly specific molecules targeted to the definite (or highly expressed) biomarkers of the tumor cells [6]. In recent decades, a variety of targeted therapeutic agents have been approved for clinical applications, or are undergoing clinical trials, that have become the standard treatment for advanced lung cancer because of their significant efficacy and safety.

Among the driver mutations of NSCLC, EGFR mutation is the most conventional driver gene in NSCLC, and approximately 10% of Caucasian, and more than 50% of Asian, NSCLC patients have been found to have sensitive EGFR mutations [7]. EGFR is reported as a subtype of the erythroblastosis oncogene B (ErbB)/human epidermal growth factor receptor (HER) family, which is also named ErbB1 (EGFR/HER1) [8,9]. As a transmembrane tyrosine kinase receptor, activated EGFR was reported to facilitate signal transduction in critical pathways of tumorigenesis [10,11]. Since the 1990s, various drug development strategies for the inhibition of overactivated EGFR have been carried out. The EGFR-targeted drugs for NSCLC that are available on the market are classified into two major categories: EGFR monoclonal antibody drugs that block the binding of extracellular ligand receptors, and small molecule chemical kinase inhibitors that inhibit the intracellular ATP binding site of tyrosine kinase [12]. Currently, the approved EGFR monoclonal antibodies are nimotuzumab and necitumumab, which are applied in combination with chemotherapy drugs in the clinic [13,14,15]. Treatment with epidermal growth factor receptor tyrosine kinase inhibitors (EGFR-TKIs) is considered more convenient, and can significantly prolong the survival time of patients, compared with the expensive therapeutic regimen of antibodies. In this review, we will focus on the indications of approved inhibitors for EGFR mutation-positive advanced NSCLC, the mechanisms of drug resistance and the corresponding therapeutic strategies, as well as the principles of reasonable and precision molecular structure for the discovery of next-generation EGFR-TKIs in order to accelerate anticancer drug discovery.

## 2. The Research Progress of EGFR-TKIs in NSCLC

The discovery of EGFR mutations in 2004 changed the standard treatment of NSCLC and established a new treatment mode according to the new molecular typing. The incidence rate of EGFR mutations is higher among women and nonsmokers. Interestingly, EGFR mutations appear widely in Asian populations [16]. The Prospective Analysis of Oncogenic Driver Mutations and Environmental Factors (PIONEER) study analyzed the tumors of 1482 patients with adenocarcinoma in mainland China, Hong Kong, Taiwan, and four other countries in Asia, including India, the Philippines, Thailand, and Vietnam, and the incidence of EGFR mutations in advanced lung adenocarcinoma was 51.4% [17]. Although EGFR mutations are more common in women and nonsmokers, they are also present in men and in 37% of regular smokers.

EGFR mutations are classified as: classical sensitive mutations (EGFR exon 19 deletion (ex19del) mutation and EGFR exon 21 p.L858R (L858R) mutation); EGFR nonclassical mutations (approximately 10%); and EGFR exon 20 insertion (ex20ins) mutations (approximately 7%) [18]. The IPASS study first proved that the first-line use of EGFR-TKIs in patients with EGFR-sensitive mutations can notably increase the objective response rate (ORR), and significantly prolong progression-free survival (PFS), compared with the results obtained from chemotherapy. Therefore, the first-line use of EGFR-TKIs has become the standard for advanced NSCLC patients with sensitizing EGFR mutations. The US Food and Drug Administration (FDA) has approved a series of EGFR-TKIs, including gefitinib, erlotinib, afatinib, dacomitinib, and osimertinib, as the first-line treatment of advanced lung cancer with EGFR mutations [19]. In China, icotinib, almonertinib, and alftinib, as original drugs, have been approved by the National Medical Products Administration (NMPA) for the treatment of NSCLC with sensitizing EGFR mutations. In addition, olmutinib was launched in South Korea as a second-line treatment for patients with advanced or metastatic NSCLC with EGFR exon 20 p.T790M (T790M) mutation positivity in 2016 (Table 1) [20].

Compared with the first-generation reversible TKIs on the market, second-generation irreversible EGFR inhibitors possess more binding sites, but result in dose-related adverse effects. Despite the significant therapeutic effects of individualized therapy, rather than traditional chemotherapy, most NSCLC patients present drug resistance after 1–2 years of treatment with first- or second-generation TKIs. More than 50% of the resistance mechanism was related to the T790M mutation. Moreover, the application of second-generation TKIs could not defeat the resistance to first-generation agents. For this reason, third-generation TKIs were designed with the superiority of binding to EGFR-sensitive mutations and T790M mutation sites, subsequently inhibiting the tumor resistance caused by the T790M mutation. However, osimeritinib, as a third-generation EGFR inhibitor, could not break the curse of drug resistance. Although the underlying mechanisms are complicated, the results of several reported clinical trials demonstrate that drug resistance still occurs. Inhibitors for different mutation targets are emerging, and various approaches could address resistance to osimeritinib. However, it is possible to overcome the obstacle of undetected mutation targets with the development of second-generation sequencing technology.

Fourth-generation EGFR-TKIs are expected to target EGFR exon 20 p.C797S (C797S) triple mutations after resistance to third-generation inhibitors; however, none of them have been officially approved to date. In 2016, compound EAI045 was reported by *Nature* to have effectively overcome, for the first time, the C797S mutation when combined with a monoclonal antibody. JBJ-04-125-02, which was further improved from the skeleton of EAI045, can effectively solve the problem of the C797S cis mutation. In addition, TQB3804, an original candidate in China that first appeared at the American Association for Cancer Research (AACR) conference in 2019, was developed to prevent EGFR C797S cis mutations (Figure 1).

## 3. First-/Second-Generation EGFR-TKIs

### 3.1. Indications of the First-/Second-Generation EGFR-TKIs

First-generation EGFR-TKIs, such as gefitinib, erlotinib, and icotinib, are selectively targeted to EGFR tyrosine kinase, which is usually expressed in epithelial solid tumors. The common sensitive mutations of EGFR include Ex19del and L858R. For stage IV NSCLC patients with a sensitizing EGFR Ex19del/L858R mutation, gefitinib, erlotinib, or icotinib is recommended as the best first-line treatment in the clinic. Inactivation of EGFR tyrosine kinase could result in the suppression of tumor growth, metastasis and angiogenesis, and the induction of tumor cell apoptosis [42].

The second-generation small molecular inhibitors on the market are afatinib and dacomitinib (an irreversible EGFR inhibitor and a HER family inhibitor, respectively) [26,30,43]. Compared with first-generation EGFR-TKIs, second-generation inhibitors bind irreversibly to EGFR, and are effective for EGFR nonclassical mutations, as well as for common mutations [44]. Nonclassical mutations of EGFR mainly include exon 20 p.S768I (S768I), exon 21 p.L861Q (L861Q), and exon 18 p.G719X (G719X). In the analysis of the LUX 2, 3, and 6 clinical trials of afatinib versus chemotherapy, the ORR of afatinib against nonclassical mutations was 71.1%, the median PFS was 10.7 months, and the overall survival (OS) was 19.7 months [45].

EGFR ex20ins mutation is a type of mutation with high heterogeneity that is difficult to treat. Currently, the majority of EGFR ex20ins subtypes (A763_Y764insFQEA and D770delinsGY) are not sensitive to EGFR-TKIs. In patients with EGFR exon 20 mutations, the overall efficacy of afatinib was poor (ORR = 7.1%, median PFS = 2.7 months, median OS = 9.2 months) [45].

### 3.2. The Mechanism of Drug Resistance and Therapeutic Strategies

#### 3.2.1. Primary Drug Resistance

Primary drug resistance refers to the first use of EGFR-TKI treatment without response. There are many reasons for primary drug resistance to EGFR-TKIs. First, there are other EGFR mutations aside from drug-sensitive EGFR mutations (ex19del and L858R). EGFR ex20ins, or other rare mutations, may be related to EGFR-TKI resistance. Second, KRAS gene mutation can contribute to drug resistance. One study showed that KRAS gene mutations existed in approximately 25% of the tumor tissues of TKI-insensitive patients, and another randomized controlled study showed that KRAS gene mutations were one of the factors influencing the efficacy of TKIs [46,47,48,49]. Third, patient characteristics, such as the rapid inactivation of metabolism, low immune function, and decreased absorption capacity, can affect the progression of drug resistance. Currently, chemotherapy is the common therapeutic used for primary drug resistance. The application of antiangiogenic therapy or immunotherapy generally depends on the individual situation, and the expression of programmed cell death 1 ligand 1 (PD-L1) in patients, respectively.

#### 3.2.2. Acquired Drug Resistance

The definition of acquired resistance includes the following aspects: (1) previous EGFR-TKI monotherapy; (2) sensitive mutations in the EGFR gene (such as ex19del, L858R, G719X and L861Q); and (3) first-line treatment with EGFR-TKIs with clinical benefits, e.g., complete remission, partial remission, or stable disease for more than 6 months [50]. Although EGFR-TKIs are effective for EGFR-mutated NSCLC, most patients acquire resistance after 7 to 11 months of therapy. The acquired resistance mechanisms of EGFR-TKIs have been reported in detail, and include the EGFR T790M mutation, HER2 amplification [51], MET amplification, and angiogenesis induction (Figure 2) [52,53,54].

##### T790M Mutation

Drug resistance inevitably develops following treatment with first-/second-generation EGFR-TKIs. Numerous mechanisms of drug resistance have been reported, and the T790M mutation is considered the most accepted mechanism. The 790th amino acid in exon 20 of EGFR alternates, from threonine to methionine, leads to the failure of EGFR-TKI to block EGFR activation, resulting in disease progression. This type of drug resistance occurs in more than 50% of cases [55,56]. Third-generation EGFR-TKIs, represented as osimertinib, can inhibit the secondary T790M mutation [57]. AURA-3 is a phase III clinical study designed to target the secondary EGFR T790M mutation in advanced NSCLC after first-line EGFR-TKI resistance, in which patients were treated with osimertinib- or platinum-based dual drug therapy. The study showed that osimertinib treatment significantly increased the ORR (71% vs. 31%) and the median PFS (10.1 months vs. 4.4 months) [58].

##### MET Gene Amplification and Overexpression

Some research has reported that resistance to EGFR-TKIs in NSCLC patients could be attributed to the amplification and overexpression of the MET gene. Approximately 20% of patients with acquired resistance to first-/second-generation EGFR-TKIs exhibit amplification of the MET gene [59]. Therefore, the combination of EGFR-TKIs and c-Met inhibitors, such as crizotinib, has become a new method for overcoming acquired drug resistance. To address the acquired resistance mechanism to EGFR-TKIs, other new targeted agents, such as shr-a1403, a conjugate of anti-met monoclonal antibody and microtubule inhibitor, are being researched and developed. Through the high affinity for tumor cell-specific antigens, the drug is transported to tumor lesions with a high expression of c-MET. After the antibody binds to the tumor cells, the microtubule inhibitor is released into the cells and exerts a tumor-killing effect [60].

##### HER2 Amplification

The HER2 mutation occurs in approximately 2–4% of NSCLC cases, most commonly in lung adenocarcinoma [52]. In NSCLC, HER2 oncogenic amplification occurs in approximately 3% of cases without EGFR-TKI treatment, and accounts for approximately 10% of cases with EGFR-TKI resistance. Case series studies and early clinical trials have shown that NSCLC patients with HER2 mutations may respond to trastuzumab-based treatment [61,62]. Patients with secondary HER2 amplification after EGFR-TKI treatment were still encouraged to participate in the clinical trials of anti-HER2 treatment because of the lack of inclusion criteria.

##### Inducing Angiogenesis

Numerous studies have reported that vascular endothelial growth factor (VEGF) and basic fibroblast growth factor (bFGF) are overproduced in different ways in solid tumors. These substances promote tumor angiogenesis and tumor metastasis, leading to a poor prognosis in NSCLC patients. The VEGF pathway has become a critical target for lung cancer therapy. As a powerful angiogenic factor, VEGF is expressed in many types of tumors. Studies have shown that some downstream factors of the EGFR pathway, such as mitogen-activated protein kinase (MAPK), and phosphatidylinositol-3-kinase (PI3K), can adjust the high expression of VEGF, resulting in tumor progression and metastasis, and acquired EGFR-TKI resistance.

The most common antiangiogenics applied in the clinic for advanced NSCLC patients include: the monoclonal antibodies of VEGF, such as bevacizumab and ramucirumab; small molecule vascular endothelial growth factor receptor 2 (VEGFR-2); and the TKI apatinib and small molecule multitarget TKIs, such as anlotinib. The voluminous prospective RCT clinical studies with large scales, listed in Table 2, indicated that the combination of bevacizumab and first- or second-generation TKIs can significantly prolong PFS and/or OS for NSCLC patients [63]. The ramucirumab monoclonal antibody is another antiangiogenic approved for the treatment of locally advanced or metastatic NSCLC, as its targets block the binding of VEGF and VEGFR2, and subsequently inhibit angiogenesis and the migration of the tumor. The clinical benefits of the combined application of ramucirumab and TKIs have been confirmed by numerous RCT studies (Table 2) [64]. Moreover, another two antiangiogenics, anlotinib and apatinib, independently developed in China, have been investigated for combined application with TKIs. For NSCLC patients with EGFR and VEGF mutations, the combined application of EGFR inhibitors and antiangiogenics can yield additional clinical benefits with the desired safety.

## 4. Third-Generation EGFR-TKI

### 4.1. Indications of Third-Generation EGFR-TKIs

A report in the *PNAS* in 2008 clarified the mechanism of T790M resistance and suggested three potential strategies to overcome T790M resistance [65]. Among them is a covalent inhibition strategy that selectively inhibits the T790M mutation of EGFR, which creates a novel core skeleton using aminopyrimidine that is different from the drug skeleton design of the first-generation inhibitor based on 4-phenylaminoquinazoline [66]. Therefore, the third-generation EGFR inhibitor, osimertinib, which specifically confers T790M mutation resistance, was designed and was successfully able to combat EGFR inhibitor resistance in approximately 50% of cases [57,67].

The third generation of EGFR-TKIs is represented by osimertinib, a specific and irreversible EGFR inhibitor. It can inhibit the T790M resistance mutation of EGFR, and the colony-stimulating factor (CSF)/plasma drug concentration ratio is also higher than that of first-/second-generation EGFR-TKIs [68]. AURA-3 is a phase III clinical study designed for the secondary EGFR T790M mutation in advanced NSCLC after first-generation TKI resistance, in which patients treated with osimertinib presented significantly increased ORR (71% vs. 31%), and PFS (10.1 months vs. 4.4 months), compared with those treated with a platinum-based dual drug treatment. The AURA-17 study further proved the efficacy of osimertinib in the East Asian population. According to the follow-up data updated by the European Society for Medical Oncology (ESMO) in 2017, osimertinib achieved a 63% ORR, and a median PFS of 9.7 months. Moreover, the results of the FLAURA phase III study displayed superiority in prolonged PFS and OS (PFS: 18.9 months vs. 10.2 months; OS: 38.6 months vs. 31.8 months) for advanced or metastatic NSCLC patients with first-line osimertinib treatment, compared with first-generation EGFR-TKI (gefitinib, erlotinib) treatment [34,35]. These well-known studies have confirmed that osimertinib has good efficacy in the first-line treatment of patients with sensitizing EGFR mutations, or in the second-line treatment of patients with T790M mutations.

Currently, almonertinib, a third-generation EGFR-TKI, was approved by NMPA (China) in 2020 for the treatment of advanced NSCLC patients with EGFR-TKI resistance progression after treatment and EGFR T790M mutation positivity. The indication was approved based on the results of the APOLLO study, a single arm, phase II clinical study that included a total of 244 patients [69]. The ORR (primary endpoint) was 68.9%, the disease control rate (DCR) was 93.4%, and the median PFS was 12.3 months. For patients with central nervous system (CNS) metastasis at baseline, the ORR and DCR of the confirmed CNS were 60.9% and 91.3%, respectively, and the median PFS was 10.8 months. Another third-generation EGFR-TKI approved by NMPA in 2021, furmonertinib, showed similar efficacy, with an ORR of 74.1%, a DCR of 93.6%, and a median PFS of 9.6 months. In a phase IIb study, 220 NSCLC patients who were T790M positive after EGFR-TKI treatment were enrolled. In patients with measurable CNS lesions, the ORR in terms of CNS lesions was 65.5%, the DCR was 100.0%, and the median PFS was 11 months [70].

### 4.2. The Mechanism of Drug Resistance and Therapeutic Strategies

The resistance mechanisms of osimertinib were classified as primary resistance and acquired resistance. The acquired resistance mechanisms were reported to be EGFR-dependent drug resistance, which was induced by remutation of the EGFR gene (on-target mutation), EGFR-independent drug resistance induced by other gene mutations (off-target mutation), and pathological transformation, such as transformation into small cell lung cancer (SCLC) and lung squamous cell carcinoma (SqCC).

#### 4.2.1. Primary Resistance

Currently, few related studies on the mechanism of primary resistance to third-generation EGFR-TKIs have been reported. The possible mechanisms include BIM deletion polymorphisms and EGFR 20 exon insertion mutations. These resistance mechanisms lead to the poor objective response rate of EGFR-TKIs in different generations.

##### BIM Deletion Polymorphism

BIM is a member of the B-cell lymphoma-2 (Bcl-2) family and a proapoptotic molecule. BIM deletion polymorphisms exist in approximately 21% of East Asians, but not in African and European populations [71]. Tanimoto et al. found that EGFR mutant lung cancer cells with BIM deletion polymorphisms were resistant to third-generation EGFR-TKIs. Recently, a case of NSCLC with EGFR L858R and T790M mutations was reported, and the effect of osimertinib was poor. BIM deletion polymorphisms were found in peripheral blood gene sequencing, and no other drug-resistant mutations were found. These results suggest that BIM deletion polymorphisms may be used as biomarkers to forecast the efficacy of osimertinib in NSCLC patients [71].

##### Insertion Mutations of EGFR Exon 20

The incidence of EGFR ex20ins mutations in NSCLC patients is approximately 3%, accounting for 10–12% of all EGFR mutations [72]. They can block the binding of EGFR-TKIs to EGFR target sites, leading to primary drug resistance. An in vitro study showed that cell lines stably expressing EGFR 20 exon insertion mutations were resistant to third-generation EGFR-TKIs [73]. These results suggest that the insertion mutation in exon 20 of EGFR may be one of the mechanisms of primary drug resistance for third-generation EGFR-TKIs. Currently, phase II clinical trials of osimertinib for NSCLC with EGFR ex20ins mutations are being recruited (NCT03414814 and NCT03191149).

#### 4.2.2. Acquired Resistance

Acquired drug resistance refers to the process by which tumor cells sensitive to previous treatments evade the influence of new drugs by changing their own metabolic pathways after contact with the third EGFR-TKI. The acquired resistance mechanism for third-generation EGFR-TKIs has been studied in more depth and detail and can be classified as EGFR-dependent resistance and EGFR-independent resistance (Figure 3) [74]. The EGFR-dependent mechanism of resistance includes EGFR remutation, T790M reduction or deletion, and EGFR amplification. Meanwhile, the EGFR-independent mechanism of resistance is more complicated and includes: the amplification of MET and HER 2; the mutation of RAS and BRAF; histological transformation; epithelial-mesenchymal transition (EMT); oncogene fusion; activation of the PI3K/serine/threonine protein kinase (Akt)/mammalian target of the rapamycin (mTOR) pathway; phosphatase and tensin homology (*PTEN*) deletion; abnormality of the fibroblast growth factor receptor (FGFR) signaling pathway; and the induction of angiogenesis [75]. The resistance mechanisms to osimertinib are discussed in detail in this section, and the corresponding treatment strategies following resistance are listed in Table 3.

##### EGFR-Dependent Resistance Mechanisms and Treatment Strategies

EGFR Remutation

EGFR remutation includes the EGFR C797S mutation, and other rare mutations, including the exon 18 p.L718 mutation, the exon 20 p.G796 mutation, and the exon 20 p.L792 mutation. The missense mutation at site 797 of exon 20 of EGFR, in which serine replaces a cysteine, is called the C797S mutation. The mutation is located in the tyrosine kinase region of EGFR, which leaves third-generation EGFR-TKIs unable to form covalent bonds in the adenosine triphosphate (ATP)-binding domain, resulting in the loss of the EGFR pathway blockade by the EGFR-TKIs. Drug resistance mechanisms have been reported to differ between first- and second-line treatments of NSCLC. As the first-line treatment of osimertinib, the frequency of the C797S mutation was approximately 7%, which was lower than that of MET amplification [76]. However, nearly 14% of the patients with progression after the second-line use of osimertinib had the C797S mutation, making it the most common mutation [77]. T790M and C797S had different drug resistance phenotypes and different follow-up treatments based on the affected alleles [78,79]. The trans structures of T790M and C797S, located in different alleles, presented resistance to third-generation TKIs, but sensitivity to the combination of first- and third-generation TKIs [80]. In contrast, when T790M and C797S were present on the same allele, the combination or single drug application of first- and third-generation EGFR-TKIs was ineffective, while fourth-generation EGFR-TKIs may be effective for this mutation. An in vitro experiment in an NSCLC mouse model (L858R/T790M/C797S triple mutations) showed that EAI045 (fourth-generation EGFR-TKI), combined with cetuximab, could effectively prevent EGFR dimerization. However, the efficacy and adverse effects of this combined treatment still need to be further investigated [81]. In addition, brigatinib (an ALK inhibitor) has been reported to inhibit EGFR L858R/T790M/C797S triple mutations [82].

T790M Deletion

All patients enrolled in the AURA-3 study had the T790M mutation after first-line treatment with first- or second-generation EGFR-TKIs, and 49% (36/73) of them had the T790M deletion after resistance to osimertinib [77]. In the studies of Le and Oxnard, T790M deletion was observed in 53% (21/40) and 68% (28/41) of drug-resistant patients, respectively [83,84]. T790M deletion may be the result of third-generation EGFR-TKI treatment and may also be one of the causes of drug resistance, which is related to the heterogeneity of tumors in the process of tumor progression. In patients with T790M retention, the resistance mechanisms are usually related to the C797S mutation or the abnormal activation of a compensatory pathway, while patients with the T790M deletion often show different resistance mechanisms, most of which are independent of the EGFR signaling pathway [85]. Repeated detection of the T790M mutation in the course of treatment is helpful for studying the mechanism of resistance to third-generation EGFR-TKIs and determining the follow-up treatment strategy. Niederst et al. suggest that first-generation EGFR-TKIs may play a role again when the T790M mutation is deleted [86]. When the T790M mutation is absent and not accompanied by other mutations, cytotoxic drugs may be appropriate for treatment.

EGFR Amplification

Nukaga et al. found an amplified EGFR gene in a rociletinib-resistant PC-9 cell line [87]. Piotrowska et al. reported that 25% (3/12) of rociletinib-resistant patients had EGFR amplification [88]. One patient with osimertinib resistance had EGFR amplification during treatment, and the amplification level was parallel to the disease stage, according to imaging findings [89]. Le et al. found that 8 (19%) out of 42 patients with osimertinib resistance had EGFR amplification [83]. EGFR gene amplification is more common in patients after rociletinib treatment, and the mechanism of drug resistance may be that EGFR gene amplification leads to a relatively low concentration of TKIs that is insufficient for inducing effects. These results suggest that clinicians can dynamically detect changes in EGF during treatment and, when patients are informed, determine whether there is drug resistance and adjust the treatment plan at the appropriate time.

##### EGFR-Independent Drug Resistance Mechanism and Treatment Strategies

Amplification of MET and HER2

The amplification of MET and HER2, detected in patients after treatment with first- and third-generation EGFR-TKIs, was reported to have a significant role in the mechanism of drug resistance. The AURA-3 study showed that 19% (14/73) of the patients had MET amplification, which was the second most common mechanism of drug resistance after the C797S mutation, and 5% (4/73) of the patients presented HER2 amplification. In vitro and in vivo experiments showed that these therapeutic regimens (crizotinib, T-DM1 alone or combined with osimertinib, osimertinib combined with MET inhibitor PF02341066 or SGX523, and the combination of capmatinib and afatinib) were modestly effective in inhibiting MET amplification [90,91,92,93]. In addition, the mid-term results of the phase IB TATTON study, published at the AACR annual meeting in 2019, showed that the combination of osimertinib and savolitinib showed acceptable safety, and preliminary antitumor efficacy in patients with MET-amplified EGFR-mutant NSCLC treated with third-generation EGFR-TKIs [94]. A phase II clinical trial called SAVANNAH is in progress. The study indicated that trastuzumab may not be effective for lung cancer patients with HER2 mutations; however, afatinib combined with cetuximab could delay the disease progression caused by HER2 amplification [95].

Mutation of RAS and BRAF

The mutation, or abnormal activation, of Ras/Raf/MEK/ERK, which is an important downstream pathway of EGFR, can lead to EGFR-TKI resistance. Eberlein et al. found that the combination of osimertinib and selumetinib (MEK inhibitor) can delay or prevent drug resistance [96]. The FLAURA study showed that BRAF mutations accounted for 3% of the resistance mechanism [76]. Another study demonstrated that osimertinib combined with encorafenib (LGX818), a BRAF V600E inhibitor, could inhibit the clonogenesis of drug-resistant cells; however, the efficacy of LGX818 alone was poor [97].

Histological Transformation

Marcoux et al. reported that 19 patients with NSCLC developed SCLC transformation after third-generation EGFR-TKI treatment, and this change was characterized by Rb1, TP53, and phosphatidylinositol-4,5-bisphosphate 3-kinase catalytic subunit alpha (*PIK3CA*) mutations [98]. The median transformation time was 17.8 months, and the median overall survival (mOS) time after transformation was 10.9 months. Lin et al. detected the circulating tumor DNA (ctDNA) of 40 patients resistant to osimertinib, in which two patients had SCLC transformation, and one had squamous cell carcinoma (SCC) transformation [99]. Roca et al. summarized the cases of lung SCC transformation in patients after EGFR-TKI treatment. The analysis showed that the median time to onset of SCC was 11.5 months, while the median survival time after the identification of SCC was 3.5 months [100]. This study also summarized 16 cases of SCC transformation after third-generation TKI treatment reported to date, 82% of whom were female, with a median transformation time of 11.5 months. All patients maintained the basic EGFR mutation. Among all patients (not sorted by treatment), the mOS time after transformation was 3.5 months [100]. The transformation of SCC after TKI resistance has not yet been observed in vitro, which indicates that the tumor microenvironment may play a potential role in this transformation process.

Epithelial-Mesenchymal Transition

EMT has been proven to be one of the resistance mechanisms to EGFR-TKIs by many studies. At the molecular level, it is mainly characterized by the decrease or disappearance of E-cadherin in epithelial cells and the increase in vimentin as a stromal marker. Weng et al. found that JMF3086 (a dual inhibitor of histone deacetylase and 3-hydroxy-3-methylglutaryl coenzyme A reductase) can reverse the sensitivity of EMT to osimertinib [101]. In addition, mesenchymal phenotype cells are sensitive to some cytotoxic agents, such as cisplatin, gemcitabine, etoposide, and vinorelbine.

Oncogene Fusion

The AURA-3 and FLAURA studies have shown that oncogene fusion may be one of the resistance mechanisms to osimertinib that include: transforming growth factor receptor (TGFR)–transforming acidic coiled coil protein 3 (TACC3); neurotrophic tyrosine receptor kinase1 (NTRK1)–thrombopoietin mimetic peptide 3 (TMP3); RET–excision repair cross complementing 1 (ERC1); spectrin beta non-erythrocytic 1 (SPTBN1)–ALK; coiled-coil domain containing 6-rearrange during transfection (CCDC6)–RET; and echinoderm microtubule-associated protein-like-4 (EML4)–ALK gene fusion. Piotrowska et al. showed that the combination of osimertinib and BLU-667 (RET inhibitor) can overcome the resistance of CCDC6–RET fusion to a certain extent [85]. Offifin et al. noted that when EML4–ALK gene fusion occurs, the combination of osimertinib and clotrimazole can delay disease progression [102].

Activation of the PI3K/Akt/mTOR Pathway

Phosphatidylinositol-4,5-bisphosphate 3-kinase catalytic subunit alpha (PIK3CA) is one of the driving genes of lung adenocarcinoma. Its mutation can promote tumor invasion and increase the activity of its downstream kinase PI3Ks. In the AURA-3 study, the incidence of PIK3CA amplification in patients resistant to osimertinib was 4% (3/73), and 1% (1/73) of patients had the PIK3CA E545K mutation [77]. In the FLAURA study, the PIK3CA mutations in first-line patients treated with osimertinib included E453K (1%), E545K (4%), and H1047R (1%) [76]. It is speculated that PIK3CA mutation or amplification can increase the activity of PI3Ks and activate various downstream kinases. This results in the activation of the PI3K/Akt/mTOR pathway, thus uncoupling upstream EGFR phosphorylation, which may be one of the resistance mechanisms of third-generation EGFR-TKIs. PPARγ agonists can overcome this resistance mechanism.

PTEN Deletion

PTEN is a tumor suppressor gene. The protein encoded by PTEN has lipid phosphatase and protein phosphatase activity, so it can play a dual antitumor role. PTEN is a key factor in many signaling pathways in the body. If the PTEN gene is mutated, deleted, or downregulated, its antitumor activity may decrease, or even be lost. To et al. suggested that a peroxisome proliferator-activated receptor gamma (PPARγ) agonist could increase the expression of PTEN, inactivate serine/threonine protein kinase (Akt), and induce autophagy in PTEN-deficient cells, thus enhancing the sensitivity to gefitinib [103]. According to the above studies, it is speculated that PPARγ agonists can reduce the occurrence of third-generation EGFR-TKI resistance.

Abnormity of the FGFR Signaling Pathway

FGFR is also a transmembrane tyrosine kinase receptor. Kim et al. found that FGFR1 amplification and fibroblast growth factor 2 (FGF2) mRNA increased in patients with resistance to osimertinib, indicating that the FGFR2-FGFR1 autocrine loop may be related to drug resistance [104]. Two patients with the T790M mutation progressed after treatment with osimertinib and naquotinib, and the FGFR3–TACC3 fusion mutation was detected by ctDNA. In the AURA-3 study, the probability of FGFR3–TACC3 fusion was 1% (1/73) [77]. It is suggested that abnormalities in the FGFR signaling pathway may be the acquired resistance mechanisms of third-generation EGFR-TKIs. Selective FGFR1 inhibitors (PD173074 and BGJ398) are effective against this resistance mechanism. However, FGF2 supplementation resulted in resistance to osimertinib in EGFR-mutated NSCLC cells.

Induction of Angiogenesis

Angiogenesis is one of the ten characteristics of malignant tumors [105]. The signaling pathway, mediated by the combination of VEGF and VEGF receptors, regulates the proliferation, survival, and migration of vascular endothelial cells, which eventually leads to angiogenesis [106,107]. The occurrence, development, and metastasis of tumors depend on angiogenesis, and anti-VEGF therapy can effectively inhibit tumor growth and tissue metastasis [108]. Some clinical research data have reported that osimertinib, combined with antiangiogenic therapy, can improve PFS and OS in cancer patients (Table 4) [109]. For instance, bevacizumab is one of the most widely used antiangiogenics. The clinical trials of osimertinib combined with bevacizumab are actively being carried out in the United States, China, Japan, and Europe. According to the existing experimental data, the combination of antiangiogenic drugs, such as bevacizumab, after resistance to osimertinib can improve the PFS and OS of patients, compared to that of osimertinib used alone [110]. Another advanced NSCLC patient was found to be making progress after eight months of a second-line therapy of osimertinib, but the primary tumor and metastasis were reduced after three weeks of therapy with apatinib [111]. Furthermore, a phase Ib clinical trial of osimertinib combined with ramucirumab for lung cancer patients with EGFR mutations has been completed in Japan, and the safety for this combination treatment has been approved.

### 4.3. Acquired Resistance Mechanisms following First- and Second-Line Osimertinib

Although the research data indicated the superiority of third-generation EGFR-TKIs, drug resistance is an inevitable problem, and the mechanisms of acquired resistance to the first- and second-line application of osimertinib vary. The AURA and FLAURA studies have focused on the mechanism of osimertinib resistance, and the corresponding resistance mechanisms are different for second- and first-line osimertinib (Figure 4). Generally, the resistance mechanisms of first- and second-line treatment are similar; however, the candidate resistance mechanism after second-line osimertinib is relatively complex. Among 91 patients who were resistant to first-line osimertinib in the FLAURA trial, the most conventional resistance mechanisms were MET amplification (15%) and EGFR-C797X mutation (7%). Others included PIK3CA (7%), HER2 amplification/mutation (3%), SPTBN1-ALK (1%), BRAF V600E (3%), KRAS (3%), and cell cycle gene alteration. No T790M mutation was found in first-line osimertinib, and neither transformation to SCLC nor SqCC was found [76]. In the AURA3 study, the gene analysis of 73 drug-resistant patients to second-line osimertinib showed that the common mechanisms of resistance included EGFR acquired mutation (21%) and MET amplification (19%), where C797S was the major EGFR acquired mutation (15%). In addition, cell cycle gene alterations (12%), HER2 amplification (5%), PIK3CA (5%), oncogene fusion (4%), and BRAF V600E (3%) were also found [77]. EGFR C797S mutations were detected in cis with the T790M mutation in patients. In addition, the T790M deletion occurred in 49% of patients, which would lead to the resistance mechanism of bypass activation. The overall mutation of second-line osimertinib resistance is more complicated, and the proportion of combined mutations after the second-line treatment was higher than that after the first-line treatment (19% vs. 14%). MET amplification and C797S mutation accounted for the largest percentage of the resistance mechanism, after both first- and second-line osimertinib (Figure 5) [112,113]. Hence, follow-up therapeutic strategies should be focused on these two gene mutations.

## 5. Exploration of Fourth-Generation EGFR-TKIs for Targeting Triple Mutations

Although third-generation EGFR-TKIs are approved as first- or second-line treatment for T790M-positive advanced NSCLC patients, patients can also develop resistance to osimertinib after 9–13 months [114]. Resistance mechanisms to third-generation inhibitors have been reported, including the EGFR C797S and *EGFR* exon p.18 L718Q (L718Q) mutations, as well as and amplification of HER2, MET, and BRAF [115,116]. Among them, the C797s mutation is the most important resistance mechanism to the third-generation inhibitor osimertinib, which prevents the formation of covalent bonds with third-generation inhibitors, affects the covalent binding of thiol groups in cysteine, and subsequently weakens the binding affinity of TKIs to EGFR [117,118]. Moreover, ex19del/T790M/C797S mutations are more common in patients with resistance to third-generation osimertinib. Although the EGFR ex19del/C797S and L858R/C797S double mutations are sensitive to first- and second-generation TKIs, and NSCLC patients with trans mutations of ex19del/T790M/C797S and L858R/T790M/C797S can be treated with osimertinib combined with first-generation TKIs, the approved EGFR-TKIs still failed to address the resistance caused by the EGFR L858R/T790M/cis-C797S mutations (Figure 6). Therefore, novel EGFR inhibitors that can effectively inhibit the EGFR ex19del/T790M/cis-C797S mutations are urgently needed [119]. A series of fourth-generation inhibitors with various binding modes have been recently reported [120].

### 5.1. EAI045

In a mechanistic study, it was stated that the ectopic inhibition of EGFR kinase is expected to overcome T790M mutation-related resistance. The design of compound EAI001 is based on the ectopic inhibition strategy of EGFR kinase. After preliminary improvement, EAI045 was obtained [81].

EAI045 was designed as an EGFR-TKI that can overcome C797S and T790M mutation-related resistance. It mainly changes the conformation of the enzyme molecule by binding to the allosteric site to inhibit the enzymatic reaction and reduce the resistance of EGFR-TKIs. To explore the therapeutic effect of EAI045, the strategy of EAI045 combined with cetuximab was applied in a mouse model with the L858R/T790M/C797S triple mutation. EAI045 could highly and selectively inhibit the proliferation of tumor cells with wild-type EGFR and the L858R/T790M mutation, and tumor volumes were significantly reduced after treatment [121]. These results indicate that EAI045 combined with cetuximab could effectively solve C797S mutation-related drug resistance, with potential benefit for patients. EAI045 is the first reported inhibitor with definite efficacy against the new generation of EGFR-resistant L858R/T790M/C797S triple mutations, and its skeletal structure is completely different from that of first- and third-generation compounds; therefore, it is known as the first global fourth-generation EGFR inhibitor, and the first global EGFR ectopic inhibitor [122].

However, EAI045 is not effective when used alone. EAI045 should be combined with cetuximab, and the limited therapeutic effect was only presented for those with the L858R/T790M/C797S triple mutation, not for those with the ex19del/T790M/C797S triple mutation, which indicates that the chemical structure of fourth-generation EGFR-TKIs still needs further improvement.

### 5.2. JBJ-04-125-02

The critical improvement in the molecular design of JBJ-04-125-02 is that the inhibition of EGFR L858R/T790M is significantly enhanced compared with that of EAI045 [81]. Although this compound is an ectopic inhibitor that must be used in combination with cetuximab for maximal benefit, it shows an excellent inhibitory effect compared with EAI045, even if used alone. Another important breakthrough is that JBJ-04-125-02 can be combined with osimertinib to enhance the efficacy of EGFR inhibition. The new generation of drugs targeting resistance-conferring mutations is more difficult to develop than previous generations, suggesting their value for clinical application [123].

However, JBJ-04-125-02 cannot inhibit ex19del/, ex19del/T790M, or ex19del//T790M/C797S. JBJ-04-125-02 did not show a good inhibitory effect on EGFR with the ex19del//T790M/C797S mutation in patients after the use of the third-generation drug osimertinib [123]. Therefore, this new drug research and development achievement is beneficial to patients who have developed drug resistance.

### 5.3. TQB3804

According to the official website of global clinical trials, the fourth-generation EGFR-TKI TQB3804 of ChiaTai TianQing has officially entered the phase I clinical research stage in China. This result suggests that patients with EGFR mutations in NSCLC will be able improve their disease status following resistance to osimertinib. Moreover, a preclinical study indicated that TQB3804 displayed a cytotoxic effect on the T790M double mutation after first- or second-generation TKI resistance. The IC50 values of TQB3804 for ex19del//T790M/C797S, L858/T790M/C797S, ex19del//T790M and L858/T790M mutations were 0.46, 0.13, 0.26, and 0.19 nm, respectively, suggesting a good inhibitory effect on tumor cells with triple and double EGFR mutations after first-, second-, and third-generation TKI resistance [124].

### 5.4. Brigatinib

Brigatinib was obtained by activity screening from a series of pyrimidine derivatives containing phosphorus designed by introducing dimethylphosphine oxide in the skeleton of the 2,4-diarylaminopyrimidineon, which consisted of dimethoxy groups with a U-shaped conformation around a bianiline scaffold [125]. Clinical research data showed that the combination of brigatinib and EGFR antibody could address resistance to the third-generation TKI osimertinib, which could inhibit ALK with the L1196M mutation, and EGFR with the T790M mutation [82]. Brigatinib was reported not only to have a certain effect on T790M double mutant NSCLC, but also to have activity on ex19del//T790M/C797S triple mutant cells, and ex19del//T790M double mutant cells. Therefore, brigatinib is considered a fourth-generation EGFR-TKI for treatment after resistance to osimertinib in NSCLC because it overcomes the mutation resistance of C797S. The standard dose of brigatinib yielded a response rate of 67% in patients with brain metastasis from lung cancer, and the PFS was 18.4 months for those with midbrain lesions.

### 5.5. BLU-945

According to preclinical data published on ESMO in 2020, and AARC in 2021, BLU-945, as the latest fourth-generation EGFR-TKI, is reported to have satisfactory cytotoxicity to EGFR double-mutant or triple-mutant (T790M/C797S or ex19del/T790M/C797S) cell lines rather than first- or third-generation TKIs. The results of the antitumor investigation in triple mutant tumor-bearing mice show that tumor regression was observed after treatment with BLU-945 only. Moreover, the mutant tumors of mice shrank rapidly until they completely disappeared after the combined application of BLU-945 and osimertinib [126,127].

## 6. Conclusions and Perspectives

With the development of biomedicine, patients with advanced NSCLC can truly benefit from precision medicine. Most patients can choose targeted therapy with few adverse effects as the first-line therapeutic regimen. Compared with standard traditional chemotherapy, EGFR-TKIs, as the first-line treatment for sensitive EGFR mutations, can prolong PFS, improve quality of life, and reduce the severe adverse effects related to therapeutics. Thus, they have become the primary treatment option for patients with advanced NSCLC. In recent decades, first-, second-, and third-generation EGFR-TKIs have been launched on the market with widespread clinical applications. Meanwhile, basic research and clinical trials of fourth-generation TKIs are also in progress. However, the occurrence and development of tumors refer to complex genetic mutations and signaling pathways, and the drug resistance of molecular targeted agents seems to be inevitable.

As a clinical therapeutic regimen for EGFR-mutant NSCLC, third-generation osimertinib has been proven to inhibit the novel T790M mutation of first- and second-generation TKIs. Fourth-generation compounds were initially designed to combat C797S mutation-mediated resistance to osimertinib. However, the inhibitory efficacy of osimertinib on L858R was significantly reduced after C797S mutation. Hence, the structure-based drug design of fourth-generation drugs also takes into account two common triple mutations. Over time, rare resistance mechanisms will be gradually unveiled. If the current research on drug discovery relies on updating the structural design of molecules on the basis of ascertainable mutations, the treatment cost for patients in the future would be beyond the acceptable budget, which is not in alignment with the original intention for drug discovery. Currently, the combination of osimertinib with other kinase inhibitors or antiangiogenics has been considered a promising therapeutic regimen in the clinic for acquired resistance. It is foreseeable that varying mechanisms of resistance to osimertinib may arise after combined application with other anticancer drugs, and whether the incidence of C797S mutation is affected following combination therapies remains to be determined. These are vital problems demanding prompt solutions in the discovery of next-generation targeted agents, and the management of individualized therapies based on precision medicine.

The emergence of drug resistance is a gradual process. Currently, the detection of mutated EGFR genes for acquired resistance is conducted by tissue or blood biopsy after disease progression, and the last-line treatment option is consequently considered. PET/CT imaging technology based on targeted molecular probes has been developed and applied in preclinical research to conveniently monitor mutated genes during therapy in a timely manner [128]. This innovative technology could provide diagnostic data and evidence for individualized clinical therapeutic regimens. However, the design of precise selective and sensitive targeted molecular probes is the key technology for the prediction of genetic mutations, which is extremely reliant upon comprehensive research and screening for structure-activity relationships [129].

With recent translational research on the biological mechanism of NSCLC progression, new pharmacotherapy approaches for protein kinase mutants continue to be developed. In addition, to protein kinase inhibitors and monoclonal antibodies, targeted protein degradation is an emerging therapeutic strategy in anticancer drug discovery. In June 2021, C4 Therapeutics reported a protein degradation agent targeting EFGR mutation. CFT8919, as a mutant selective degrader, was developed to target the degraded EGFR L858R mutation. Meanwhile, CFT8919 was reported to be active against resistant mutations, such as EGFR T790M and C797S, but to have low activity against EGFR*^WT^*, which indicated its potential clinical value for NSCLC patients.

Targeted therapy for lung cancer is a creative strategy with an exciting perspective that benefits more than half of patients. Future research trends and strategic processes for the targeted therapy of NSCLC require novel drug development with high efficiency to overcome drug resistance, combined therapeutic options to benefit the long-term survival of NSCLC patients, clinical therapeutic regimens grounded on the characteristics and genotypes of patients, and individualized whole process management schemes on the basis of precision medicine. In summary, with the development of drug discoveries and the innovation of therapeutic strategies in the future, lung cancer is expected to become a curable chronic disease.

## Figures and Tables

**Figure 1 pharmaceutics-13-01500-f001:**
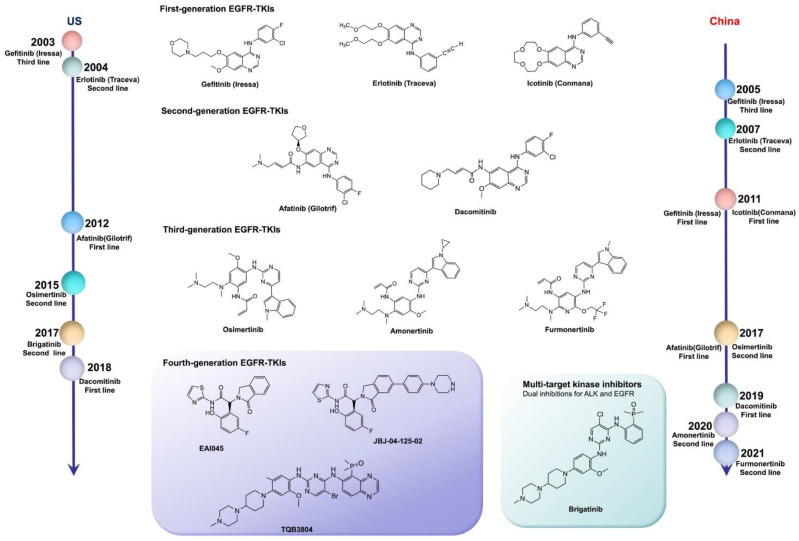
The timeline of EGFR-TKI discovery for targeted therapy of NSCLC in the U. S. and in China, respectively. EGFR-TKIs: epidermal growth factor receptor kinase inhibitors; ALK: anaplastic lymphoma kinase.

**Figure 2 pharmaceutics-13-01500-f002:**
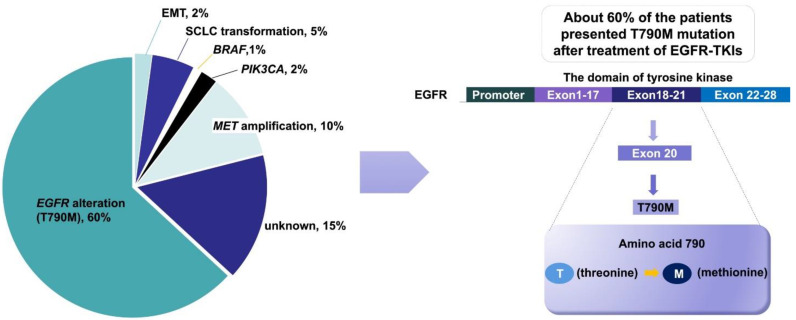
Mechanism and mutation frequency of acquired resistance to EGFR-TKIs. Reprinted with permission from ref. [52]. Copyright 2018 Springer Nature.

**Figure 3 pharmaceutics-13-01500-f003:**
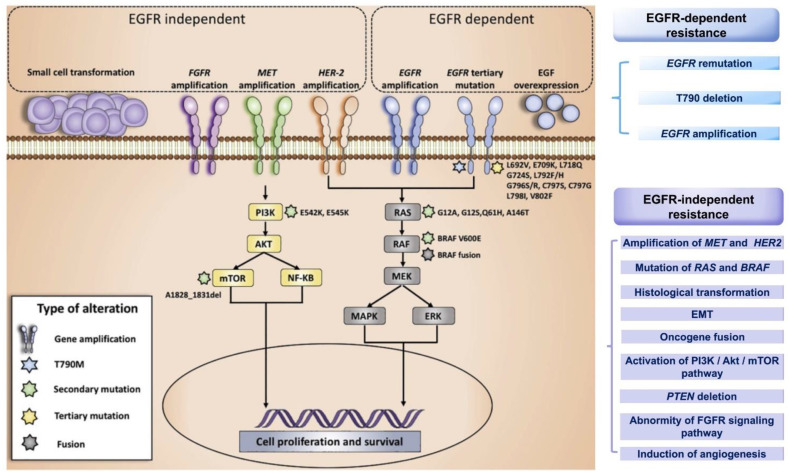
The acquired resistance mechanisms to third-generation EGFR-TKIs can be classified as EGFR-dependent resistance and EGFR-independent resistance. The EGFR-dependent mechanism: EGFR re mutation, T790M deletion and EGFR amplification; The EGFR-independent mechanism: amplification of MET and HER2, mutation of RAS and BRAF, histological transformation, EMT, oncogene fusion, activation of the PI3K/Akt/mTOR pathway, *PTEN* deletion, abnormity of the FGFR signaling pathway, and the induction of angiogenesis. (MET: mitogen activation kinase; EMT: epithelial mesenchymal transition; *PTEN*: phosphatase and tensin homology; FGFR: fibroblast growth factor receptor; ERK: extracellular regulated protein kinases). Reprinted with permission from ref. [74]. Copyright 2018 Elsevier Science.

**Figure 4 pharmaceutics-13-01500-f004:**
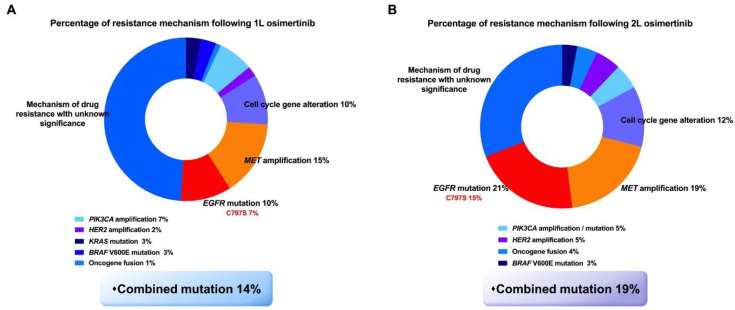
Difference of acquired resistance mechanisms between first-line (**A**) and second-line (**B**) treatment of third-generation TKI osimertinib. Adapted from [76,77], published by Future Medicine, 2018 and Elsevier, 2018.

**Figure 5 pharmaceutics-13-01500-f005:**
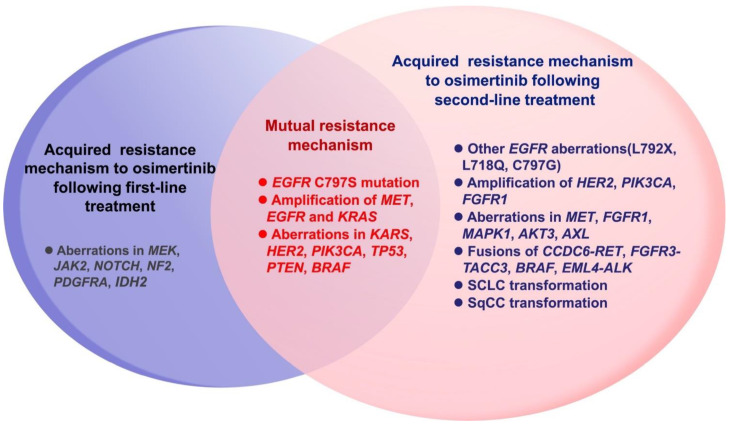
Differences in drug resistance mechanisms between first-line and second-line after the third-generation EGFR-TKIs treatment.

**Figure 6 pharmaceutics-13-01500-f006:**
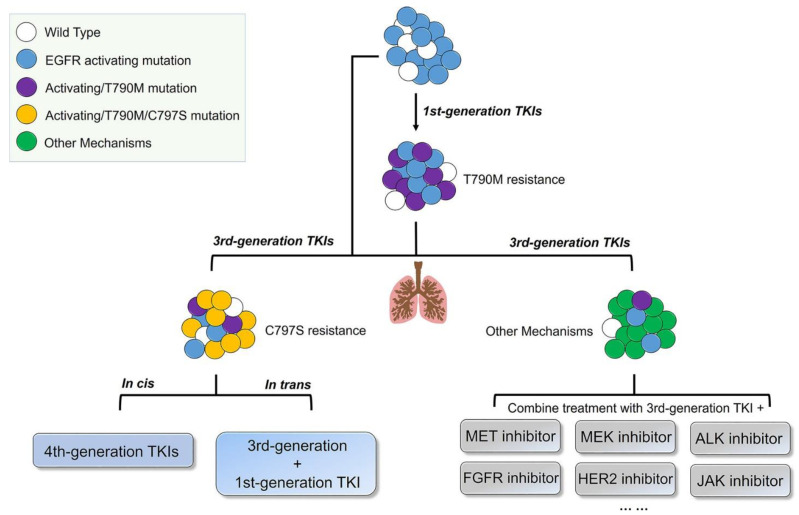
Comprehensive post-therapeutic strategies of drug resistance for EGFR mutant NSCLC-targeted therapy. The colored ball represents acquired resistance mechanism, which indicates the heterogeneity of tumor (JAK: Janus-activated kinase). Reprinted with permission from ref. [115]. Copyright 2018 ACS American Chemical Society.

**Table 1 pharmaceutics-13-01500-t001:** Clinical progress of first-/second-/third-generation EGFR-TKIs.

Product Name	Phase	Company	Trial ID	Study Design	mPFS (Months)	HR	mOS (Months)	ORR	DCR	Control Group	Ref.
Gefitinib	III	AstraZeneca	NCT00322452	RCT	9.5 vs. 6.3	0.48	21.6 vs. 21.9	71.2% vs. 47.3%	/	Carboplatin + Paclitaxel	[21]
Icotinib	III	China	NCT01040780	RCT	4.5 vs. 3.4	0.84	13.3 vs. 13.9	27.6% vs. 27.2%	75.4% vs. 74.9%	gefitinib	[22]
Icotinib	III	China	NCT01719536	RCT	11.2 vs. 6.9	0.61	30.5 vs. 32.1	64.8% vs. 33.85%	/	Cisplatin + pemetrexed	[23]
Erlotinib	III	Roche	NCT00446225	RCT	9.7 vs. 5.2	0.37	19.3 vs. 19.5	55% vs. 11%	78% vs. 66%	platinum-containing dual drugs	[24]
Erlotinib	III	Roche	NCT00874419	RCT	13.7 vs. 4.6	0.16	22.8 vs. 27.2	83% vs. 36%	96% vs. 82%	gemcitabine/carboplatin	[25]
Afatinib	III	Boehringer Ingelheim	NCT01121393	RCT	11.1 vs. 6.9	0.58	28.2 vs. 28.2	56% vs. 24%	90% vs. 82%	cisplatin/pemetrexed	[26]
Afatinib	III	Boehringer Ingelheim	NCT01121393	RCT	11.0 vs. 5.6	0.28	23.1 vs. 23.2	66.9% vs. 23.0%	92.6% vs. 76.2%	cisplatin/gemcitabine	[27]
Afatinib	II	Boehringer Ingelheim	NCT01466660	RCT	11 vs. 10.9	0.74	27.9 vs. 24.5	70% vs. 56%	91.3% vs. 87.4%	gefitinib	[28]
Afatinib	III	Boehringer Ingelheim	NCT01523587	RCT	2.6 vs. 1.9	0.81	7.9 vs. 6.8	5.5% vs. 2.8%	50.5% vs. 39.5%	erlotinib	[29]
Dacomitinib	III	Pfizer	NCT00446225	RCT	14.7 vs. 9.2	0.59	34.1 vs. 26.8	74.9% vs. 71.6%	/	gefitinib	[30]
Gefitinib	III	AstraZeneca	WJTOG3405	RCT	9.2 vs. 6.3	0.49	34.9 vs. 37.3	62.1% vs. 32.2%	93.1% vs. 78%	cisplatin/docetaxel	[31]
Gefitinib	III	AstraZeneca	NEJ0020376	RCT	10.8 vs. 5.4	0.32	27.7 vs. 26.6	73.7% vs. 30.7%	/	carboplatin/paclitaxel	[32]
Erlotinib	III	Roche	NCT01342965	RCT	11.0 vs. 5.5	0.34	26.3 vs. 25.5	62.7% vs. 33.6%	89.1% vs. 76.6%	cisplatin/gemcitabine	[33]
Osimertinib	III	AstraZeneca	NCT02296125	RCT	18.9 vs. 10.2	0.42	38.6 vs. 31.8	80% vs. 76%	97% vs. 92%	gefitinib/erlotinib	[34,35]
Osimertinib	III	AstraZeneca	NCT02151981	RCT	11.7 vs. 5.6	0.32	/	70% vs. 31%	93% vs. 63%	platinum + pemetrexed	[36]
Osimertinib	II	AstraZeneca	NCT03424759	Single arm studies	8.2	/	/	50%	89%	/	[37]
Osimertinib	II	AstraZeneca	NCT02228369	Single arm studies	8.6	/	11	41%	/	/	[38]
Almonertinib	II	China	NCT02981108	Single arm studies	12.3	/	/	68.9%	93.4%	/	[39]
Olmutinib	II	South Korea	NCT02485652	Single arm studies	9.4	/	19.7	51.9%	81.4%	/	[40]
Furmonertinib	II	China	NCT03452592	Single arm studies	9.6	/	/	74.1%	93.6%	/	[41]

RCT: randomized controlled trial; mPES: median progression-free survival; HR: hazard ratio; mOS: median overall survival; ORR: objective response rate; DCR: disease control rate.

**Table 2 pharmaceutics-13-01500-t002:** Clinical trials of the combination of EGFR-TKIs and antiangiogenics after resistance to first-/second-generation TKIs.

Drug Name	Phase	Clinical Trial Registration Number	Country	Study Design	mPFS	HR	OS	ORR	DCR	Research Design	Status
Anlotinib	III	NCT04797806	China	RCT *	/	/	/	/	/	Anlotinib + Icotinib vs. Icotinib	Recruiting
Bevacizumab	II	JapicCTI-111390 and JapicCTI-142569	Japan	RCT	16.4 m vs. 9.8 m	0.52	47 m vs. 47.4 m	69% vs. 64%	99% vs. 88%	Bevacizumab + Erlotinib vs. Erlotinib	NR *
Bevacizumab	III	UMIN000017069	Japan	RCT	16.9 m vs. 13.3 m	0.61	50.7 m vs. 46.2 m	72% vs. 66%	/	Bevacizumab + Erlotinib vs. Erlotinib	No longer recruiting
Bevacizumab	II	/	/	/	14.4 m	/	/	0.738	0.976	Bevacizumab + Gefitinib	* NR
Bevacizumab	/	/	China	RCT	8.35 m vs. 4.61 m		11.27 m vs. 7.05 m	41.5% vs. 21.3%	82.9% vs. 63.4%	Bevacizumab + Icotinib vs. Icotinib	* NR
Bevacizumab	I/II	NCT02803203	USA	/	19 m	/	/	0.8	1	Bevacizumab + Osimertinib	Active, not recruiting
Bevacizumab	III	NCT02336451	USA	RCT	18.0 m vs. 11.3 m	0.55	/	86.3% vs. 84.7%	/	Bevacizumab + Erlotinib vs. Erlotinib	Completed
Bevacizumab	II	NCT01532089	USA	RCT	17.9 m vs. 13.5 m	0.87	/	83% vs. 81%	/	Bevacizumab + Erlotinib vs. Erlotinib	Completed
Bevacizumab	II	NCT01562028	France	/	mPFS:13.2 m; T790M(+): 16.0 m; T790M(−): 10.5 m	/	/	/	/	Bevacizumab + Erlotinib	Completed
Bevacizumab	II	NCT00531960	Australia	/	mPFS:18.4 w vs. 25 w	/	/	/	/	Bevacizumab + Erlotinib vs. Bevacizumab + chemotherapy	Completed
Bevacizumab	/	/	/	Observational study	23.9 m		45.9 m	87.70%	100.00%	Bevacizumab + afatinib	* NR
Bevacizumab	III	NCT02633189	Italy	RCT	/	/	/	/	/	Bevacizumab + Erlotinib vs. Erlotinib	Active, not recruiting
Bevacizumab	III	NCT02759614	China	RCT	/	/	/	/	/	/	Unknown status
Bevacizumab	II	NCT00367601	USA	/	3.4 m	/	5.1 m	40%	/	Bevacizumab + Erlotinib	Completed
Bevacizumab		NCT03647592	China	RWS **	/	/	/	/	/	Bevacizumab + Erlotinib/Gefitinib	Recruiting
Bevacizumab	II	NCT00354549	Switzerland	/	4.1 m	/	/	10.90%	/	Bevacizumab + Erlotinib, ** GP/GC after the PD	Completed
Bevacizumab	III	NCT00130728	USA	RCT	3.4 m vs. 1.7 m	0.62	9.3 m vs. 9.2 m		45% vs. 34%	Bevacizumab + Erlotinib vs. Erlotinib	Completed
Bevacizumab	/	NCT04575415	China	RWS	/	/	/	/	/	Bevacizumab + EGFR-TKI	Recruiting
Ramucirumab	/	/	/	RCT	17.1 m	/	/	/	/	Ramucirumab + Erlotinib	Active, not recruiting
Ramucirumab	III	NCT02411448	USA	RCT	19.4 m vs. 12.4 m	0.59	/	76% vs. 75%	95% vs. 96%	Ramucirumab + Erlotinib vs. Erlotinib+Placebo	Active, not recruiting
Ramucirumab	III	NCT02411448	USA	RCT	19.4 m vs. 12.5 m	0.636	/	/	/	Ramucirumab + Erlotinib vs. Erlotinib+Placebo	Active, not recruiting
Ramucirumab	III	NCT02411448	USA	RCT	20.6 m vs. 10.9 m	0.605	/	/	/	Ramucirumab + Erlotinib vs. Erlotinib+Placebo	Active, not recruiting
Apatinib	/	/	China	RCT	7.6 m vs. 3.0 m (second-line vs. third-line)	/	/	0.033	0.9	Apatinib + TKI	* NR
Apatinib	/	/	China	RCT	5.47 m	/	/	0.138	0.862	Apatinib + Icotinib/Erlotinib/ Gefitinib	* NR
Apatinib	/	/	China	RCT	12.1 m vs. 8.6 m	/	/	73.9% vs. 52%	95.7% vs. 92.5%	Apatinib + Icotinib vs. Icotinib	* NR
Apatinib	/	/	China	RCT	not research point	/	/	0.304	/	Apatinib + Icotinib/Erlotinib/ Gefitinib/Osimertinib	* NR
Apatinib	/	/	China	RCT	14.3 m vs. 10.3 m	/	/	76% vs. 68%	96% vs. 92%	Apatinib + Gefitinib vs. Gefitinib	* NR

* NR: not reported; ** GP: Gemcitabine + Cisplatin, GC: Gemcitabine + Carboplatin, PD: Progressive Disease. RCT: randomized controlled trials; RWS: randomized withdrawal study; mPFS: median progression-free survival; HR: hazard ratio; OS: overall survival; ORR: objective response rate; DCR: disease control rate.

**Table 3 pharmaceutics-13-01500-t003:** Mechanisms and treatment strategies for resistance to osimertinib.

Third-Generation EGFR-TKIs	Resistant Mechanisms	Countermeasures
osimertinib	*EGFR* C797S Cis mutation	The combination uses of first-and third-generation EGFR-TKIs; first-generation EGFR-TKIs;
osimertinib	*EGFR* C797S Trans mutation	Fourth-generation EGFR-TKIs
osimertinib	*EGFR* G796R	Chemotherapy (docetaxel)
osimertinib	*EGFR* G796S/D	* NR
osimertinib	*EGFR* L792H	Combination of oxitinib and cetuximab
osimertinib	*EGFR* L792F/R/Y/V/P/I/S	* NR
osimertinib	*EGFR* L798I	First-generation EGFR-TKIs; the second-generation EGFR-TKIs
osimertinib	Deletion of T790M mutation	First-generation EGFR-TKIs; cytotoxic drugs
osimertinib	*EGFR* amplification	EGFR mAbs
osimertinib	*MET* amplification	Crizotinib; trastuzumab emtansine (T-DM1); Combination of carmatinib and afatinib;
osimertinib	*HER2* mutation	Combination of alfatinib and cetuximab
osimertinib	*RAS* mutation	selumetinib
osimertinib	BRAF mutation	Enorafenib (LGX818)
osimertinib	Cell type transformation	SCLC transformation: chemotherapy (etoposide and carboplatin); SCC transformation: double drug chemotherapy with platinum;
osimertinib	EMT	JMF3086; cytotoxic drugs;
osimertinib	Oncogene fusion	RET inhibitor
osimertinib	Activation of PI3K/Akt/mTOR pathway	PPARg agonist
osimertinib	PTEN deletion	PPARγ agonist
osimertinib	FGFR amplification	FGFR1 inhibitor: PD173074, BGJ398;

EGFR: epidermal growth factor receptor; MET: mesenchymal to epithelial transition factor; HER2: human epidermal growth factor receptor 2; BRAF: v-raf murine sarcoma viral oncogene homolog B1; SCLC: small cell lung cancer; SCC: squamous cell carcinoma; EMTE: epithelial-mesenchymal transition; RET: rearrange during transfection; PTEN: phosphatase and tensin homology; FGFR: fibroblast growth factor receptor; NR: not reported.

**Table 4 pharmaceutics-13-01500-t004:** Clinical trials of the combination of EGFR-TKIs and antiangiogenics after resistance to osimertinib.

Drug Name	Phase	Clinical Trial Registration Number	Country	Study Design	mPFS	HR	OS	ORR	DCR	Research Design	Status
Bevacizumab	I/II	NCT02803203	USA	/	19 m	/	/	0.8	1	Bevacizumab + Osimertinib	Active, not recruiting
Bevacizumab	I/II	UMIN000023761	Japan	RCT	13.5 m vs. 9.4 m	/	not reach vs. 22.1 m	/	/	Bevacizumab + Osimertinib vs. Osimertinib	No longer recruiting
Bevacizumab	II	NCT03133546	Europe	RCT	/	/	/	/	/	Bevacizumab + Osimertinib vs. Osimertinib	Active, not recruiting
Bevacizumab	II	NCT04425681	China	/	/	/	/	/	/	Bevacizumab + Osimertinib	Recruiting
Bevacizumab	III	NCT04181060	USA	RCT	/	/	/	/	/	Bevacizumab + Osimertinib vs. Osimertinib	Recruiting
Bevacizumab	II	NCT02971501	USA	RCT	/	/	/	/	/	Bevacizumab + Osimertinib vs. Osimertinib	Active, not recruiting
Ramucirumab	Ib	UMIN000030164	Japan	/	9.2 m	/	/	0.83	/	Ramucirumab + Osimertinib	Main results already published
Apatinib	/	/	China	RCT	not research point	/	/	0.304	/	Apatinib + Icotinib/Erlotinib/Gefitinib/Osimertinib	* NR

* NR: not reported; RCT: randomized controlled trial; mPFS: median progression-free survival; HR: hazard ratio; OS: overall survival; ORR: objective response rate; DCR: disease control rate.

## Data Availability

Not applicable.

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
