# Peer review of "Iterative Upgrading of Small Molecular Tyrosine Kinase Inhibitors for EGFR Mutation in NSCLC: Necessity and Perspective"

_pharmaceutics, 2021, doi:10.3390/pharmaceutics13091500_

Round 1

Reviewer 1 Report

The manucript entitled "Iterative Upgrading of Small Molecular Tyrosine Kinase Inhibitors for EGFR Mutation in NSCLC: Necessity and Perspective" highlighted that the mechanisms of drug resistance and the corresponding therapeutic strategies, as well as the principle of reasonable and precision molecular structure for drug development discovery of next-generation inhibitors for EGFR, which would accelerate the anticancer drug discovery.

- The manuscript may benefit from a language revision by an English native speaker.
- The Authors should provide the expand forms for all acronyms, including gene acronyms, through the text when they first appear.
- Gene acronyms should be written in italics.
- Mutations should be reported as follow: e.g. EGFR exon 21 p.L858R.

Reviewer 2 Report

In this extensive review the authors evaluated the indications of approved inhibitors for EGFR mutation-positive advanced NSCLC, the mechanisms of drug resistance and the corresponding therapeutic strategies, as well as the principle of reasonable and precision molecular structure for drug development discovery of next generation EGFR-TKIs, whichwould accelerate the anticancer drug discovery.

The manuscript is well written and organized and offers a complete update on the role of EGFR-TKIs.

Reviewer 3 Report

This is a very nice and informative review on the clinical significance of small molecular TKIs to treat NSCLC with EGFR Mutations. The manuscript contains a lot of very interesting information, is quite up-to-date, nicely written, and presented and I believe it will be of value and very helpful not only for the readers familiar with the field of NSCLC but for all readers working in the field of anticancer drugs.

This is a review that focuses mainly on TKI drugs that target EGFR mutations.

Authors present in a concise yet informative manner the history and the development of such drugs. They discuss the roadmap from the first to the fourth generation drugs. They have included all four generation which allows to the reader not only to get enough information on the current status but also to the problems arising (like the presence of the T790M the met and HER 2 genes amplification, the to the C797S mutation etc) in treating resistant NSCLC. Importantly they analyze the steps that lead to greater resistance as therapy is applied and is easy to follow the changes that lead to the need for the discovery of new drugs. Importantly and although this review is primarily focused on EGFR mutations they also give information on the escape pathways that may developed in NSCL, which gives the reader the actual biological frame that leads to the development of résistance in NSCLC. In three nice tables they also report all the clinical trials related to these drugs while in a fourth one they have put together the problems that arise in the use of the most important third-generation TKI osimertinib in a way that aids the reader to understand clearly the problem.  It is noteworthy also to underline that authors report in a quite detailed manner all fourth generation under development drugs.

Finally, they summarize all this information in a nice conclusion and perspective parts (that needs though some fine-tuning regarding the language).

Overall this is a nice review that provides the reader with a lot of old and novel information in the field of TKIs specifically used to target resistant NSCLC but also pointing out what the actual needs are to fight this deleterious type of cancer.

Round 2

Reviewer 1 Report

I have no further comments.